# SYPL1 defines a vesicular pathway essential for sperm cytoplasmic droplet formation and male fertility

Jiali Liu [1,2,10], Louis Hermo[3,10], Deqiang Ding[1,4,10], Chao Wei[1], Jeffrey M. Mann[1], Xiaoyuan Yan[1], Ashley F. Melnick [1], Yingjie Wu[1,5], Alicia Withrow[6], Jose Cibelli[1], Rex A. Hess [7] & Chen Chen [1,8,9] ✉

The cytoplasmic droplet is a conserved dilated area of cytoplasm situated at the neck of the sperm flagellum. Viewed as residual cytoplasm inherited from late spermatids, the cytoplasmic droplet contains numerous saccular elements as its key content. However, the origin of these saccules and the function of the cytoplasmic droplet have long been speculative. Here, we identify the molecular origin of these cytoplasmic droplet components by uncovering a vesicle pathway essential for formation and sequestration of saccules within the cytoplasmic droplet. This process is governed by a transmembrane protein SYPL1 and its interaction with VAMP3. Genetic ablation of SYPL1 in mice reveals that SYPL1 dictates the formation and accumulation of saccular elements in the forming cytoplasmic droplet. Derived from the Golgi, SYPL1 vesicles are critical for segregation of key metabolic enzymes within the forming cytoplasmic droplet of late spermatids and epididymal sperm, which are required for sperm development and male fertility. Our results uncover a mechanism to actively form and segregate saccules within the cytoplasmic droplet to promote sperm fertility.

During spermiogenesis, haploid spermatids drastically change their shape (round to elongated), and form an acrosome and flagellum to fulfill specialized functions as future spermatozoa[1,2]. The cytoplasmic droplet (CD, also called Hermes body)[3] of spermatozoa is a conserved expanded area of cytoplasm found in association with the flagellum in all mammalian species but often overlooked in sperm biology[4–8]. Since its discovery in 1909[9], the function of the CD has been elusive. In early studies, the CD was regarded as a residual cytoplasm harboring degenerating ER and Golgi remnants retained by sperm[10] when the late

spermatid discards most of its other organelles and cytoplasm into residual bodies at the time of sperm release from the testis[11,12]. Viewed as a passively formed structure, the CD is maintained on the sperm flagellum as sperm traverse the epididymis. The CD is initially positioned at the neck region of sperm adjacent to its head, but moves in a peristaltic manner from the neck region to the distal end of the flagellar mid-piece where the annulus is positioned during sperm transit[13,14]. Isolated CDs detached from sperm contain ample metabolic enzymes and its presence is positively correlated with motility, leading

[1]Department of Animal Science, Michigan State University, East Lansing, MI, USA. [2]State Key Laboratory of Animal Biotech Breeding, College of Biological Sciences, China Agricultural University, Beijing, China. [3]Department of Anatomy and Cell Biology, McGill University, Montreal, Quebec, Canada. [4]Clinical and Translational Research Center of Shanghai First Maternity and Infant Hospital, Frontier Science Center for Stem Cell Research, School of Life Sciences and Technology, Tongji University, Shanghai, China. [5]College of Animal Science and Technology, China Agricultural University, Beijing, China. [6]Center for Advanced Microscopy, Michigan State University, East Lansing, MI, USA. [7]Department of Comparative Biosciences, University of Illinois, Urbana, Illinois, USA. [8]Reproductive and Developmental Sciences Program, Michigan State University, East Lansing, MI, USA. [9]Department of Obstetrics, Gynecology and Reproductive Biology, Michigan State University, Grand Rapids, MI, USA. [10]These authors contributed equally: Jiali Liu, Louis Hermo, Deqiang Ding. ✉e-mail: chen2@msu.edu

to a postulated role in sperm maturation[3,12,15–19]. However, the definitive function of the CD has been difficult to assess due to the lack of in vivo models that abolish or alter its contents.

The primary contents of the CD are numerous membranous vesicles in the form of flattened, curled or dilated saccules, morphologically resembling those of the Golgi stack[11]. These saccules congregate into the forming CD of step 16 spermatids in large quantity before sperm are released from the testis[11]. However, the derivation and molecular origin of these saccules and the underlying mechanism for their sequestration within the forming CD remain undisclosed.

Here by genetic ablation of Synaptophysin-like protein 1 (SYPL1), an integral membrane protein related to synapse vesicle protein Synaptophysin and interacting with the v-SNARE protein vesicle associated membrane protein 3 (VAMP3), we uncover a mechanism for trans Golgi network (TGN) vesicle production that is critical for the formation of saccules and their segregation within the CD, an event essential for animal fertility. SYPL1 also controls glycolytic enzyme enrichment in the CD prior to spermiation. Our study suggests that the CD is an actively formed region of cytoplasm of the sperm tail housing saccular elements designated to store important protein reserves for sperm maturation to ensure fertility.

## Results

### SYPL1 is enriched in the cytoplasmic droplets of spermatozoa

During the final steps of mouse spermiogenesis, hexokinase 1 (HK1), a key glycolytic enzyme and marker of the CD[20], exhibits a late onset of cytoplasmic expression in steps 14-15 spermatids and consequently is highly enriched in the forming CD of step 16 spermatids (Fig. 1a). To date, the mechanism for its transport and highly enriched segregation within the CD is unknown. Considering the numerous saccules within

the CD (Fig. 1b), we hypothesize a role for these membranous elements in this process.

Amongst the many proteins described by proteomic studies of isolated CDs of rat and mouse sperm, several vesicle trafficking proteins were enriched in mouse and rat CDs[3,16]. Among them, SYPL1 is a transmembrane protein with four transmembrane helices (Fig. 1c) and defines constitutive cytoplasmic transport vesicles with a diameter <100 nm in cultured cells[21]. SYPL1 was abundant in a proteomic analysis of isolated CDs of epididymal sperm[3]. However, the expression and in vivo function of SYPL1 are unknown. Western blot analysis revealed that SYPL1 is expressed in multiple mouse tissues with high expression in the testis (Fig. 1d). To explore the role of SYPL1 in male germ cells, we examined cell type-specific expression and subcellular localization of SYPL1 in the mouse testis by immunofluorescence. SYPL1 was exclusively localized in the cytoplasm of germ cells including spermatocytes (SPC), round spermatids (RS), elongated spermatids (ES), and spermatozoa (SPZ). Its expression was not detected in spermatogonia (SPG) and somatic cells such as Sertoli cells (SC) and Leydig cells (LC) (Supplementary Fig. 1). The subcellular distribution of SYPL1 in germ cells displayed a small punctate pattern in the cytoplasm, indicative of a vesicular protein expression pattern. In particular, SYPL1 was highly enriched in the CDs of testicular and epididymal sperm (Fig. 1e, f), suggesting a potential role in spermiogenesis and sperm maturation.

### SYPL1 is essential for male fertility

To understand the physiological function of SYPL1 and explore the role of SYPL1 in sperm development, we generated mice with a targeted mutation in *Sypl1* where exon 3 was replaced by a LacZ cassette (Fig. 2a). *Sypl1* homozygous mutant mice (KO) were protein null as indicated by the absence of SYPL1 in KO testes by Western blotting and

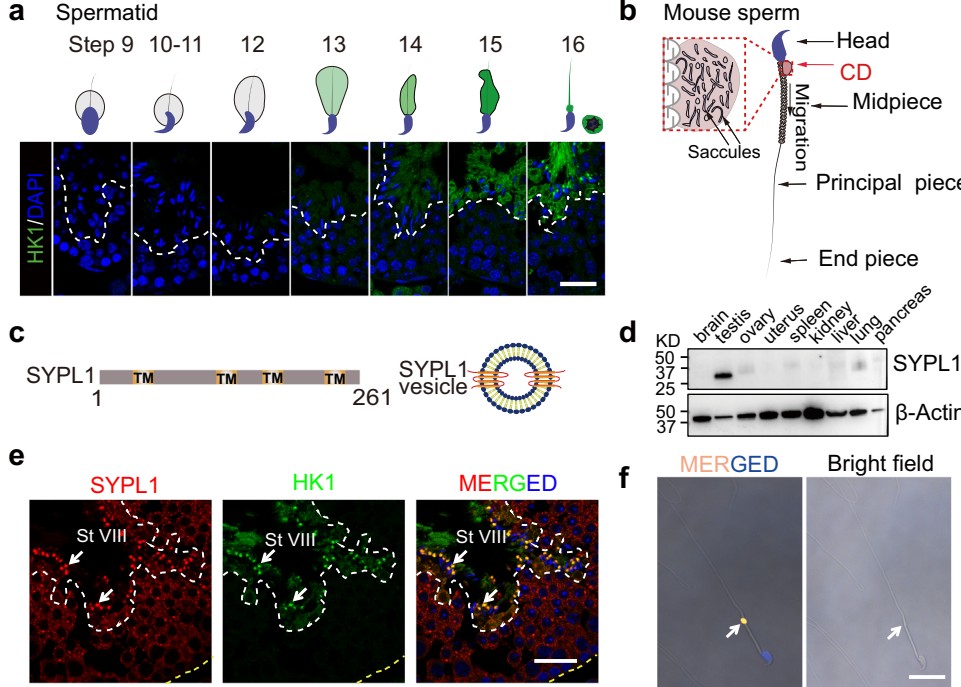

**Fig. 1 | SYPL1 is enriched in the cytoplasmic droplet. a** Immunofluorescence and corresponding schematic illustration showing the expression of HK1 in mouse step 9–16 spermatids. Scale bar, 30 μm. **b** A schematic diagram of mouse sperm bearing a CD. **c** A schematic illustration of SYPL1 protein domain architecture. Note the four transmembrane regions (orange)[21]. **d** Western blot of SYPL1 expression in multiple mouse tissues. β-Actin serves as a loading control. **e** Co-localization of SYPL1 and HK1 in WT mouse testes. White arrows indicate CDs. Open dotted white lines indicate the border between round spermatids and elongated spermatids. Open dotted yellow lines indicate the border of the seminiferous tubule. St, stage; scale bar, 30μm. **f** Immunofluorescence showing SYPL1 expression in the CD of mouse cauda epididymal sperm. The white arrow indicates SYPL1-positive signal in the CD region. Scale bar, 20 μm. Source data are provided as a Source Data file.

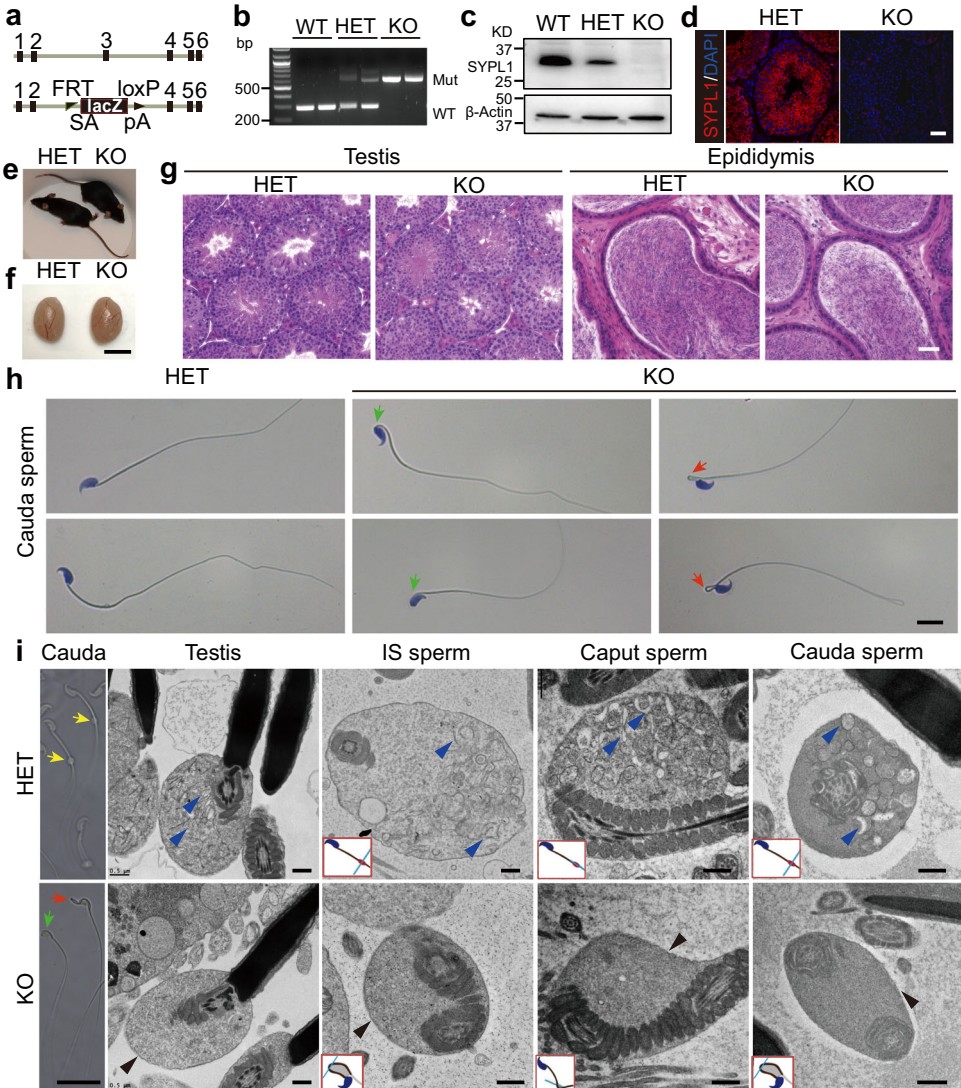

**Fig. 2 | Loss of saccular elements in the CDs of KO sperm. a** A schematic diagram of *Sypl1* WT and mutant alleles. **b** Representative results of PCR genotyping using mouse tail DNA. **c** Western blot showing the expression of SYPL1 in WT, HET, and KO testes. **d** Immunofluorescence showing SYPL1 in HET and KO testes. Scale bar, 50 μm. **e, f** Images of HET and KO mice and testes. Scale bar, 5 mm. **g** Histology of HET and KO testes and epididymides. Scale bar, 50 μm. **h** Brightfield micrographs of isolated sperm from cauda epididymides of HET and KO mice stained with hematoxylin. Abnormal morphology is indicated by green arrows (bent head) and red arrows (bent midpiece). Scale bar, 10 μm. **i** Phase contrast and transmission electron micrographs showing CD morphology and saccular elements in HET and KO sperm. Yellow arrows indicate CDs. Abnormal morphology is indicated by green (bent head) and red (bent midpiece) arrows. Scale bar, 20 μm (left panel). Blue arrowheads indicate saccular elements. Black arrowheads indicate absence of saccular elements in CDs. Small schematic diagrams indicate the cutting positions of sperm in corresponding figures. IS, initial segment; Caput, caput epididymis; Cauda, cauda epididymis. Scale bar, 0.5 μm.

immunofluorescence (Fig. 2b–d). KO mice were born at a Mendelian ratio (30:60:32), suggesting that SYPL1 is not vital for embryonic development. Adult KO mice appeared normal and healthy and were indistinguishable from wild-type (WT) or heterozygous (HET) control mice (Fig. 2e). The body and testis weights of KO mice were similar to those of littermate controls (Fig. 2e, f).

To examine whether SYPL1 deficiency caused any defect in spermatogenesis, we performed histological analysis of KO testes and epididymides. Similar to controls, all germ cell populations (spermatogonia, spermatocytes, round spermatids and elongated spermatids) appeared normal in KO testes. Numerous mature sperm could be found in KO epididymides (Fig. 2g). Epididymal sperm count revealed no significant differences between KO and control mice (Supplementary Fig. 2a). However, KO epididymal sperm showed severely impaired motility (Supplementary Fig. 2b and 2c).

Intriguingly, morphological analysis revealed that KO epididymal sperm exhibited abnormal flagellar angulation, mainly in a bent

head at the neck region of sperm or bent mid-piece phenotype (Fig. 2h and Supplementary Fig. 2f). The percentages of angulated sperm were significantly higher than that of controls in testes, caput and cauda epididymides (Supplementary Fig. 2d, e). To assess whether impaired sperm morphology and motility in KO mice affect male fertility, we tested the fertility of *Sypl1* KO male mice. Despite normal mating behavior, 8 out of 10 KO males tested were completely infertile. The remaining 2 KO males did produce only 1 litter each of reduced size during a 5-month breeding period (Supplementary Table 1). Together, these results indicate that *Sypl1* is a critical gene for sperm maturation and male fertility.

**SYPL1 determines saccule formation and segregation in the CD**
To further understand the morphological abnormality of KO sperm, we compared testicular and epididymal spermatozoa from HET and KO mice by transmission electron microscopy (TEM). Concomitant with the bent head and bent mid-piece phenotype, the expanded area

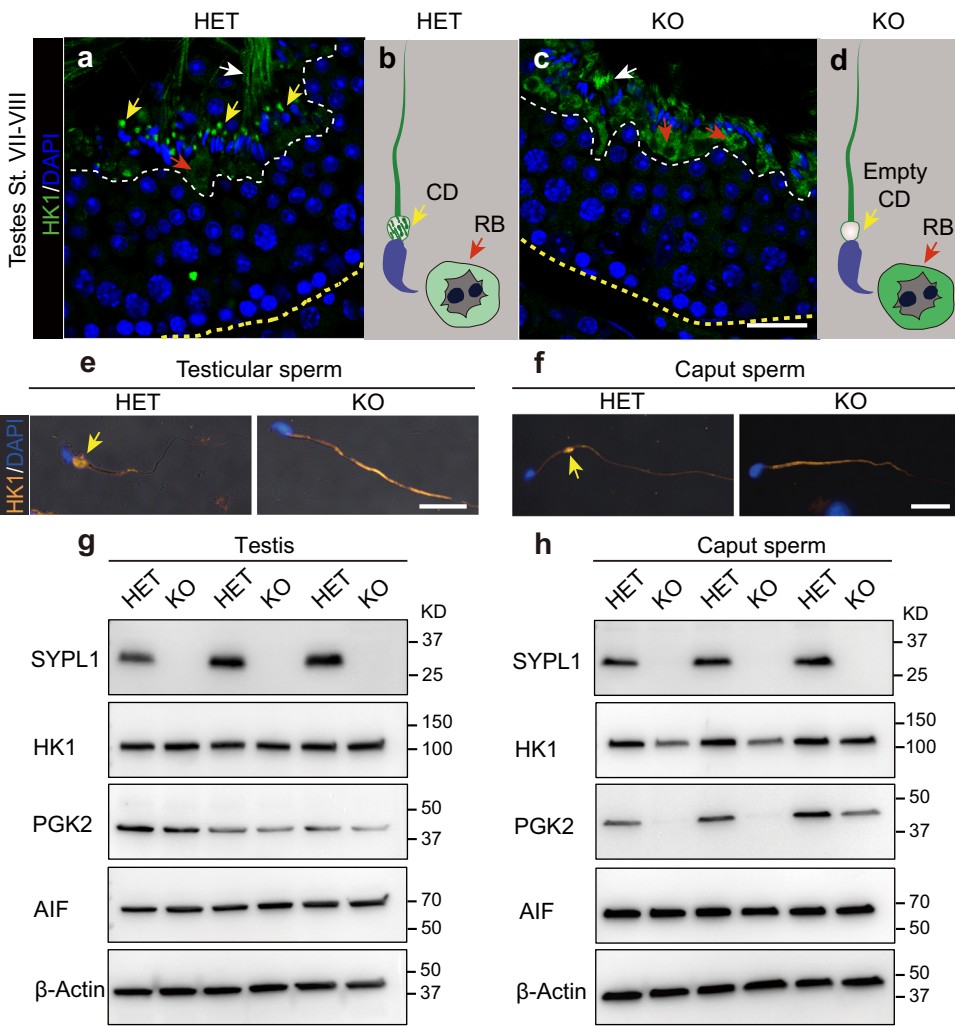

**Fig. 3 | HK1 fails to be sequestered within the CD in KO testes.**
**a–d** Immunofluorescence showing the localization of HK1 in HET and KO testes. Yellow arrows indicate CDs. Red arrows indicate residual bodies (RB). White arrows indicate sperm flagella. Corresponding schematic diagrams for a and b are shown in **b**, **d**, respectively. St, stage; Scale bar, 30 μm. **e**, **f** Immunofluorescence of HK1 in step 16 spermatids and caput epididymal sperm of HET and KO mice. Yellow arrows indicate CDs. Scale bar, 20 μm. **g**, **h** Western blots of SYPL1, HK1, PGK2, AIF in testes and caput epididymal sperm of HET and KO mice. Results from three pairs of HET and KO mice are shown. β-Actin serves as a loading control. Source data are provided as a Source Data file.

of the flagellum referred to as the CD appeared to be greatly diminished in size and devoid of saccular elements in KO epididymal sperm (Fig. 2i). The lack of a prominent CD bulge in KO sperm was also confirmed by scanning electron microscopy (Supplementary Fig. 2g) and the counting of epididymal CD-bearing sperm by phase microscopy (Supplementary Table 2).

To further delineate the defect of CD formation in KO sperm, we analyzed TEM data of testis and epididymis sections. In HET testis, step 16 spermatids harbor numerous saccules in the expanded CD area (Fig. 2i). These saccules revealed a flattened or dilated membranous shape with a straight or curved profile in the forming CD of the testis and different regions of the epididymis (Fig. 2i). Hybrid forms of saccules revealing dilated extremities and a constricted central area were also noted (e.g., Fig. 2i, testis). However, in KO step16 spermatids, very few saccules, if any, were observed in the CD region and the lack of saccules was also characteristic of KO epididymal sperm (Fig. 2i, Supplementary Table 3). Despite being devoid of saccules in the CD region, the integrity of the CD region remained intact, although diminished in size. An electron dense finely granular background was apparent in the CD of KO sperm along with occasional small round vesicles (Fig. 2i; Supplementary Fig. 3). Besides the CD defect, other structures and organelles of mutant sperm such as mitochondrial

sheath, acrosomes, nuclei and flagellar components appeared normal. Together, these data reveal the absence of saccular elements in the CD. This strongly suggests that SYPL1 is the key determinant of the formation and sequestration of the saccular elements, the most commonly seen membranous structure in the CD of animal spermatozoa.

## SYPL1 sequestrates glycolytic enzymes to the CD

To analyze whether the enrichment of CD proteins was affected by SYPL1 deficiency, we examined in situ localization of CD enriched proteins in testes of control and KO mice. In HET stage VII/VIII seminiferous tubules, HK1 was concentrated in discrete high intensity expanded foci adjacent to the neck region of late step 16 spermatids, indicative of CD localization, with weak signals in residual bodies (Fig. 3a and b). In stark contrast, no bright HK1 foci representing the CDs were observed in stage matched KO tubules. Instead, the level of HK1 in residual bodies was higher compared with HET, indicating the failure of normally expressed HK1 to be concentrated in the CD region (Fig. 3c and d).

Consistent with the loss of saccular elements in CDs of KO epididymal sperm (Fig. 2i), immunofluorescence revealed that the enrichment of HK1 to the CD region was undetectable in isolated KO testicular sperm and caput epididymal sperm (Fig. 3e, f). However, HK1 signal could be observed in caput KO sperm flagella where it appeared

as a homogeneously reactive layer along the entire mid-piece of the flagellum, suggesting that SYPL1 is selectively required for a concentrated HK1 sequestration to the CD (Fig. 3e, f).

Similar to HK1, another key glycolytic enzyme phosphoglycerate kinase 2 (PGK2) was concentrated in the CD of control mice of step 16 spermatids and epididymal sperm, but not in the CD of epididymal KO sperm where it appeared as a homogeneous weak reaction along the flagellar mid-piece (Supplementary Fig. 4). Western blot of HK1 and PGK2 in control and KO testes and caput epididymal sperm showed decreased protein levels of HK1 and PGK2 in KO epididymal sperm, but not in KO testes (Fig. 3g, h), while other non-CD expressing proteins, such as AIF and β-Actin, were not altered by SYPL1 deficiency (Fig. 3g, h). Collectively, these results indicate that the enrichment of protein components selectively to the CD is disrupted due to SYPL1 deficiency, suggesting a critical role for SYPL1 in sequestering enrichment of protein cargos within the CD.

### Golgi-derived SYPL1 localizes to the membrane of CD saccules
Since SYPL1 concentrated to the forming CD before the release of step 16 spermatids, we examined the subcellular localization of SYPL1 during spermatid development. Use of anti-SYPL1 antibody exhibited spherical immunoreactive aggregates in the cytoplasm during different steps of spermatid development (Fig. 4a). In step 8-13 spermatids the reaction appeared in several aggregates including a large prominent reactive aggregate; subsequently, SYPL1 dispersed into smaller aggregates in the cytoplasm of step 14–15 elongated spermatids. Thereafter, SYPL1 was enriched and concentrated in the forming CDs at the neck of step 16 spermatids (Fig. 4a). Such observations correspond to the gradual fragmentation of the Golgi apparatus during late steps of spermiogenesis as visualized by TEM with remnants with a Golgi profile retained in the forming CD[11,12,20,22,23].

To explore the organelle identity of the SYPL1 aggregates in spermatids, co-staining of SYPL1 with various cellular organelle markers was conducted. No overlapping signals were observed in the co-staining of SYPL1 with mitochondrial, endosomal and endoplasmic reticulum markers in elongating spermatids (Supplementary Fig. 5a). Interestingly, a prominent SYPL1 reactive aggregate was observed adjacent to cis Golgi network (CGN) markers RCAS1 (Fig. 4b) and GM130 (Supplementary Fig. 5b and 5c), and co-localized with trans Golgi network (TGN) marker Golgin97 in step 8-12 spermatids (Fig. 4a and Supplementary Fig. 5b), indicating the Golgi identity of the SYPL1 aggregate. Considering the important role of the Golgi apparatus in acrosome formation in spermatids, we performed co-staining of acrosome marker ACRV1 with SYPL1 in WT and KO testes. SYPL1 did not co-localize with ACRV1 in WT elongating spermatids and *Sypl1* deletion did not alter the formation of the acrosome (Supplementary Fig. 6).

To further explore the subcellular localization of SYPL1 in the elongated spermatids and CD, we performed immuno-electron microscopy (IEM) and showed that SYPL1 was localized on the membrane of small intracellular vesicles (putative precursor CD vesicles) in step 13–14 spermatids as well as on the membrane of the flattened or dilated straight or curved saccular elements of the forming CD of step 16 spermatids and those in epididymal sperm (Fig. 4c). Together, the specific co-localization of a TGN marker of the Golgi apparatus with SYPL1 suggests the Golgi identity of SYPL1-positive saccular elements. The presence of SYPL1 saccules in the CD further suggests a Golgi-derived, SYPL1-mediated vesicular process for saccular formation. In addition, the absence of saccules in *Sypl1* KO mice suggests a role for SYPL1 in sequestration of these saccules within the forming CD of step 16 spermatids.

### SYPL1 partners with SNARE protein VAMP3 during CD formation
Next, we investigated whether other proteins are involved in SYPL1 vesicular process during CD formation. The vesicle-associate membrane protein (VAMP) family, a class of transmembrane v-SNARE

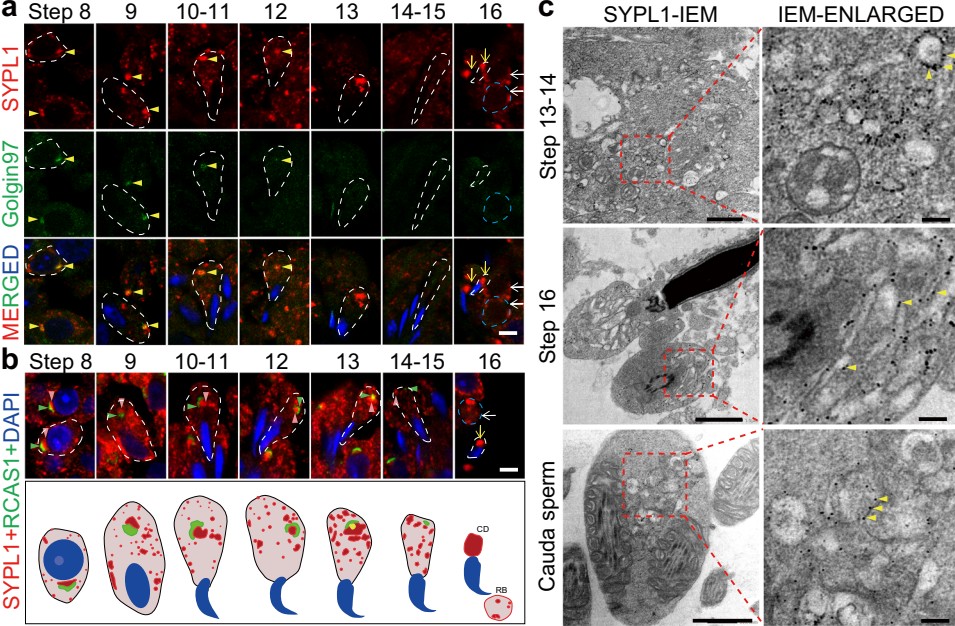

**Fig. 4 | SYPL1 is derived from trans Golgi network. a** Co-localization of SYPL1 and trans Golgi network marker Golgin97 in WT step 8–16 spermatids by immunofluorescence. Yellow arrowheads show co-localization of Golgin97 and SYPL1 in step 8–12 spermatids. Yellow arrows indicate CDs. White arrows indicate residual bodies (RB). Open dotted white lines indicate the borders of spermatids. Open dotted blue lines indicate the boarders of RB. Scale bar, 5 μm. **b** Co-localization of SYPL1 and cis Golgi network marker RCAS1 in WT step 8–16 spermatids by immunofluorescence. Pink arrowheads indicate SYPL1 positive signal and green arrowheads indicate the cis Golgi network. Yellow arrows indicate CDs and white arrows indicate RBs. A schematic diagram of the top panel is shown in the bottom panel. Scale bar, 5 μm. **c** Immuno-electron microscopy (IEM) localization of SYPL1 in WT step 13–16 spermatids and cauda sperm. SYPL1 signals are indicated by yellow arrowheads. Scale bars, 1 μm (left panel) or 200 nm (right panel).

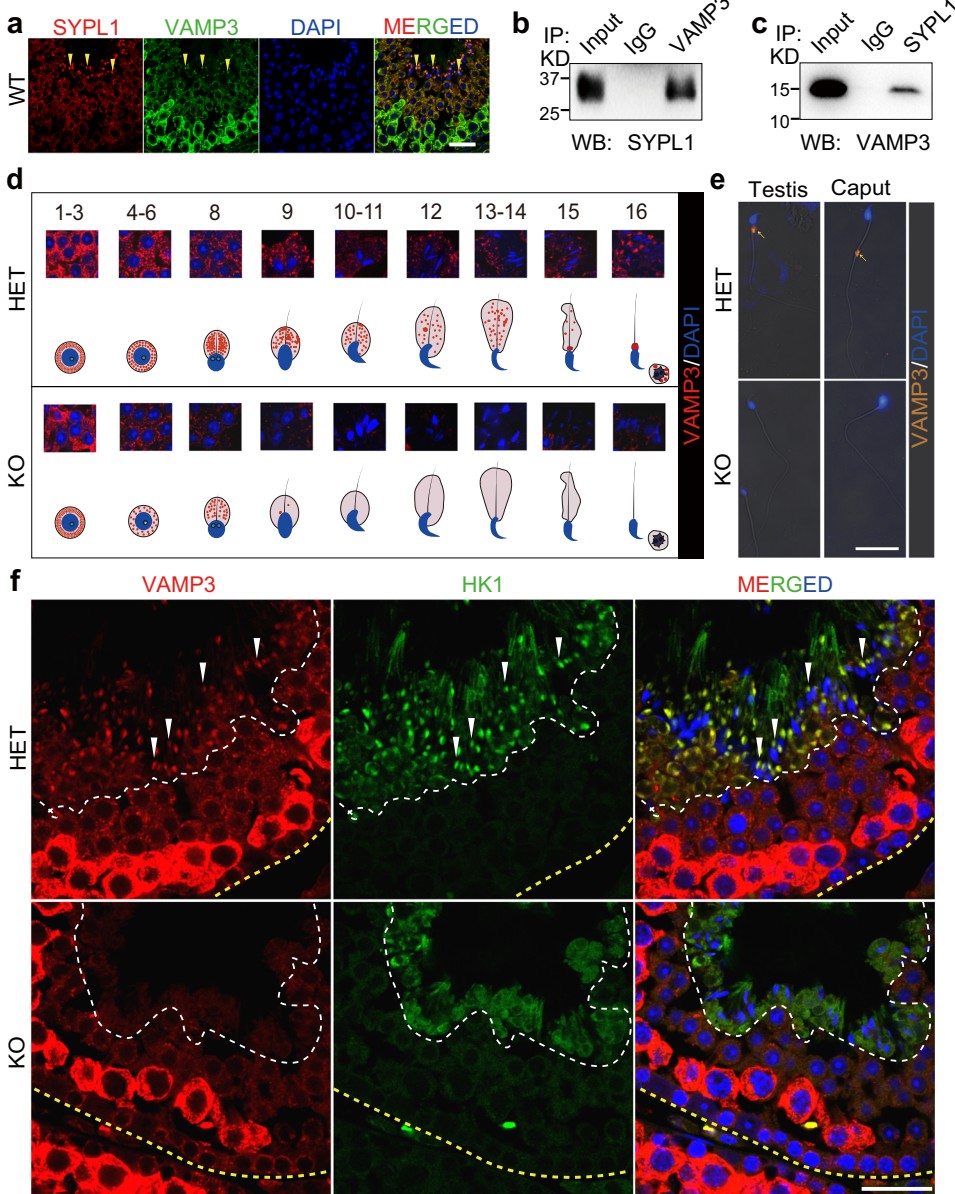

**Fig. 5 | VAMP3 is regulated by SYPL1 during CD formation.**
**a** Immunofluorescence co-localization of SYPL1 and VAMP3 in WT testes. Yellow arrowheads indicate CDs. Scale bar, 30μm. **b**, **c** Co-immunoprecipitation and Western blotting of SYPL1 and VAMP3 using WT testicular lysates.
**d** Immunofluorescence of VAMP3 in step 1–16 spermatids of HET and KO mice. A schematic diagram is shown beneath the images. **e** Immunofluorescence of VAMP3 in testicular and caput epididymal sperm of HET and KO mice. Yellow arrows indicate CDs. Scale bar, 20 μm. **f** Co-localization of VAMP3 and HK1 in Stage VIII seminiferous tubules of HET and KO testes. White arrowheads indicate CDs. Scale bar, 30 μm. Source data are provided as a Source Data file.

proteins involving vesicle trafficking, is known to associate with the Synaptophysin family proteins including Synaptophysin and SYPL1[24–26]. To explore a specific partnership between SYPL1 and the VAMP family in the context of CD saccular formation, we examined the expression and localization of the VAMP family members VAMP2, VAMP3 (also known as cellubrevin) and VAMP4 in mouse testes. While VAMP2, VAMP3 and VAMP4 are all expressed in the mouse testis (Supplementary Fig. 7a), only VAMP3 co-localized with SYPL1 and presented a similar staining pattern seen as small spherical reactive aggregates in the cytoplasm of germ cells (Fig. 5a, Supplementary Fig. 7b-d). Notably, VAMP3, but not VAMP2 or VAMP4, was enriched in CDs of step16 spermatids partnering with SYPL1 (Fig. 5a, Supplementary Figs. 7b-d). Indeed, the detection of VAMP3 in the forming CD of mouse is confirmed by immunofluorescence of rat CDs at step 19 spermatids in rat and by a proteomic analysis of the isolated rat CD[3].

To validate the interaction of SYPL1 and VAMP3 in vivo, we performed co-immunoprecipitation experiments in WT mouse testes. VAMP3 was specifically detected in SYPL1 immunoprecipitates and vice versa (Fig. 5b, c). Together, these results indicate that SYPL1 interacts with VAMP3 and may function together to regulate vesicle fusion and trafficking during the formation of saccular elements in late spermatids.

To understand the relationship between SYPL1 and VAMP3, we examined the expression changes of these two proteins during spermatid development in WT testes. Although SYPL1 and VAMP3 colocalized almost exclusively in several cytoplasmic aggregates during the course of spermatid development, one large solitary aggregate, resembling TGN, was positive only for SYPL1 but not for VAMP3 (Supplementary Fig. 7d). This interesting pattern suggests that SYPL1 could originate first from Golgi and subsequently engage with VAMP3 in the formation of co-expressing cytoplasmic vesicles.

We next examined whether there was any effect of SYPL1 deficiency on VAMP3 expression and localization. As compared to HET, VAMP3 expression commenced in early KO round spermatids but was diminished in step 11 to 15 elongating spermatids (Fig. 5d, Supplementary Fig. 8b). In step 16 KO spermatids, VAMP3 was undetectable in both the CD region and residual body (Fig. 5d, f, Supplementary Fig. 8b). The absence of VAMP3 was also confirmed in isolated testicular and epididymal KO sperm (Fig. 5e, Supplementary Fig. 8c). The reduction in protein expression was specific for VAMP3 because SYPL1 deficiency did not cause protein level changes in HK1 (Fig. 5f) suggesting a possible role of SYPL1 in controlling VAMP3 expression and/or stability. RT-PCR results showed that the mRNA level of *Vamp3* did not change in KO testes compared with HET testes, suggesting that the regulation of VAMP3 by SYPL1 is post-transcriptional (Supplementary Fig. 8a).

### SYPL1-mediated saccular formation in spermatid development

To further understand SYPL1-mediated vesicular process during CD formation in spermatids, we used TEM to trace CD vesicles in HET and KO spermatids at distinct developmental steps of spermiogenesis. In step 12 spermatids, typical Golgi structures from which SYPL1 vesicles appeared to originate were noted in both HET and KO cells (Fig. 6a, f). In step 14–15 HET spermatids, a large cluster of small spherical vesicles could be observed in the cytoplasm and on one side of the mitochondrial sheath (Fig. 6b–d). However, in steps 14–15 KO spermatids, large membrane bound vacuolar structures were evident along with a few normal-looking small vesicles (Fig. 6g–i). As HET spermatids developed to step 16, numerous flattened or dilated straight or curved saccular elements appeared in the CD region (Fig. 6e). In sharp contrast, flattened saccules were absent in the CD region of step 16 KO spermatids with only very few spherical vesicles being present (Fig. 6j).

These dramatic differences in WT and KO spermatids suggest that loss of SYPL1 leads to improper saccular formation and sequestration to the CD region. It is suggested that in the absence of SYPL1, VAMP3 function is altered, as already discussed above, leading to the uncontrolled vesicular fusion resulting in large vacuolar structures in the cytoplasm of late spermatids and in the "empty" CD phenotype. While late spermatids do show endocytic vesicles in control animals, they are small and not abundant[27] and no evidence exists for a phagocytic function. Hence these large vacuolar structures would not qualify as

endocytic or phagocytic structures. Their exclusion from the forming CD may be due to their abnormal shape and/or incorrect genetic profile, an area for future experimentation.

## Discussion

Since the discovery of the CD over 100 years ago, the molecular nature of the saccular elements of the CD remains unknown, which precludes the assessment of their function. Here we uncovered a cellular pathway that dictates the formation of the saccular elements of the CD via a Golgi mediated vesicular process involving SYPL1 and VAMP3 during late steps of spermiogenesis. The presence of these saccules within the CD is also crucial for the sequestration of protein cargos such as HK1 to the forming CD of step 16 spermatids and that of epididymal sperm where it is highly concentrated. Taken together the data provide the molecular insight into the origin of the saccular elements and their sequestration within the CD, while also suggesting a possible explanation for their significance by localizing protein cargos to this site (Fig. 7).

The CD's membranous components have been speculated to represent degenerating Golgi elements based on morphology[10], and the fragmentation of the Golgi ribbon leading to formation of individual Golgi saccules that eventually concentrate in the forming CD of late spermatids[11]. Here we show that SYPL1 colocalizes with TGN and appears to be instrumental in formation of small vesicles that fuse together to form the saccular elements as well as sequester them within the CD of late spermatids. This supports the Golgi origin for the saccules contained within the CD[11], and provide the earliest molecular signature for a SYPL1-derived vesicular origin for the saccules of the CD in spermatids. We found that VAMP3 is post-transcriptionally regulated by SYPL1 and its expression is diminished upon SYPL1 deficiency. We speculate that VAMP3 may facilitate vesicle fusion by the stabilization of SYPL1, resembling a mechanism used by SYN-VAMP2 in the nervous system[24,26,28]. Thus, the fusion of Golgi-derived vesicles in late spermatids appears to be dependent on SYPL1 and its interaction with VAMP3 and as such give rise to saccular elements that are present in the CD. Remarkably, In *Sypl1* KO mice, large vacuoles were noted in the cytoplasm suggesting that an uncontrolled fusion process occurs in the absence of SYPL1 which also results in altered expression of VAMP3, leading to malformation of the proper saccular element formation and sequestration to the CD. Concomitantly, the finding of not

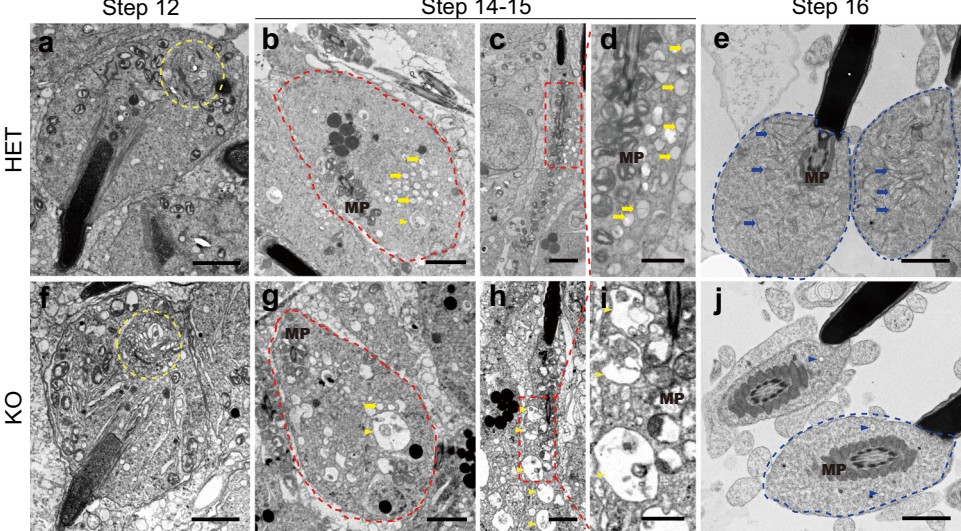

**Fig. 6 | SYPL1 deficiency leads to absence of saccular elements in the forming CD. a–e** Transmission electron micrographs (TEM) showing the ultrastructure of step 12–16 spermatids of HET mice. **f–j** TEM showing the ultrastructure of step 12–16 spermatids of KO mice. Yellow dotted circles indicate Golgi apparatus. Red dotted circles indicate transverse cut of step 14–15 spermatids cytoplasm. **d, i** are enlarged areas of boxed areas in **c** and **h**. Yellow arrows indicate vesicles and yellow arrowheads indicate large vacuoles. Blue dotted circles indicate CDs. Blue arrows indicate saccular elements in the CD of HET spermatozoa. Blue arrowheads indicate small vesicles in the CD region of KO spermatozoa. MP, midpiece. Scale bars, 2 μm in **a–d**, **f–i**, and 1 μm in **e, j**.

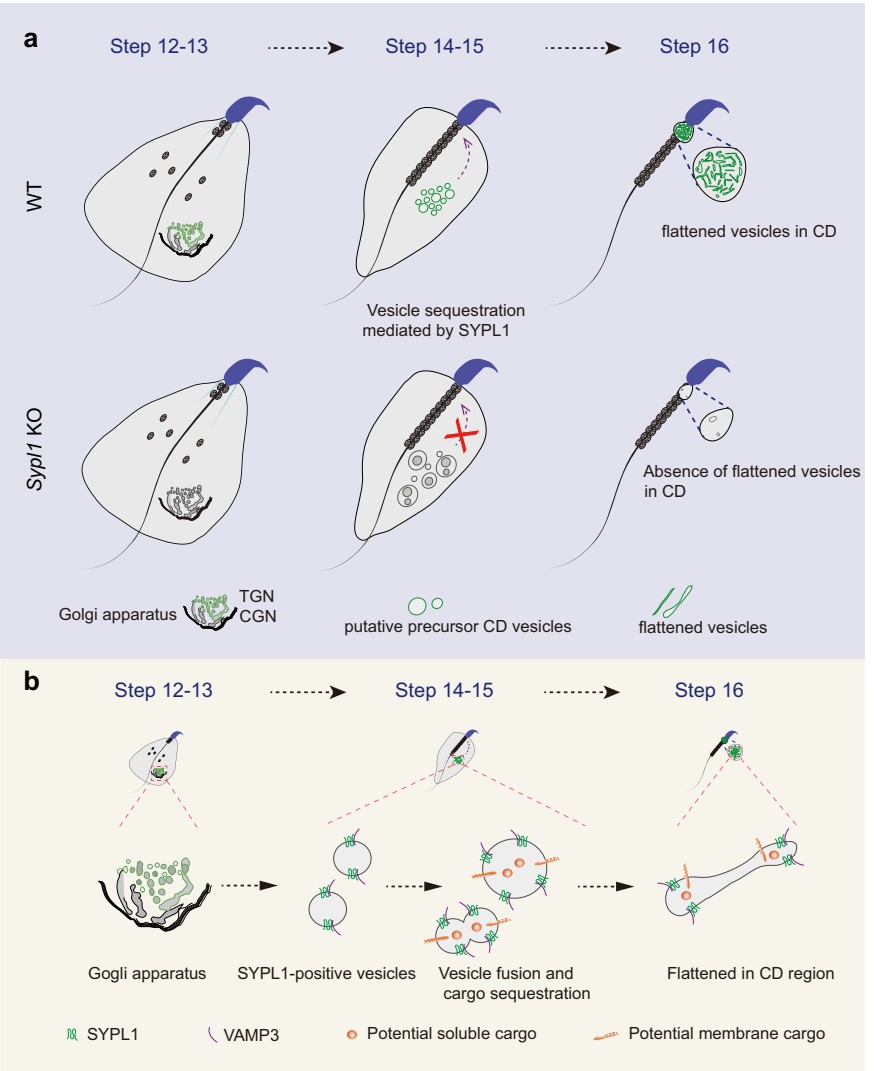

**Fig. 7 | A proposed model of the role of SYPL1 in CD formation. a** A schematic diagram depicting the course of CD formation in WT and KO spermatids. TGN-derived SYPL1 vesicles in late spermatids actively form the saccular elements that are sequestered within the forming CD of WT sperm. *Sypl1* deletion results in the absence of flattened saccules in the CD region of late spermatids. SYPL1-positive vesicles are labeled in green. **b** Schematic enlargement of SYPL1-positive vesicles to form saccules of the CD and sequester potential cargos to this site.

only flattened and dilated saccules but hybrid saccules suggests that a fusion process is occurring for saccular formation.

The precise function of VAMP3 in CD formation is unknown. Unlike *Sypl1* KO males, *Vamp3* KO male mice are fertile[28,29], reflecting a functional difference between SYPL1 and VAMP3. This can be attributed to the differences in their molecular functions. As a v-SNARE protein, VAMP3's best characterized function is to regulate vesicle fusion in various cellular contexts[30–34]. By contrast, SYPL1, as a membrane residential protein, may control multiple other proteins to regulate vesicle formation, transport and fusion simultaneously[21]. Indeed, we observed the expression of SYPL1 preceded that of VAMP3 in the TGN region of the Golgi apparatus in the WT elongating spermatids, further implicating its role in initial vesicle formation (Supplementary Fig. 7). The precise roles of SYPL1 and VAMP3 in saccular formation and sequestration warrant further biochemical and genetic investigation.

Isolated CDs contain numerous metabolic enzymes, and the presence of CDs is correlated with the potential of sperm motility development[3,16]. We observed that glycolytic enzymes HK1 and PGK2, though expressed at normal levels, failed to accumulate in the forming CD regions of *Sypl1* KO step 16 spermatids and epididymal sperm.

Intriguingly, while HK1 and PGK2 are absent in the putative CD regions of KO sperm, both are detectable in the cytoplasm along the mid-piece of KO sperm flagella and residual bodies. This highlights the importance of the presence of SYPL1 saccules in the CD to concentrate these glycolytic enzymes at this site and that they are specialized to segregate specific protein cargos to the CD, which differs from the sperm intra-flagellar transport pathway[35,36].

SYPL1 deficiency leads to an "empty CD" phenotype. This is distinctive from other described abnormalities in CD development[37–42]. The absence of glycolytic enzymes in the empty CD of *Sypl1* KO spermatozoa correlates with their motility defects. This coincides with the finding that epididymal spermatozoa that display initial and progressive motility predominantly possess CDs, whereas spermatozoa without CDs are rarely motile[15]. Given the observed remodeling of Golgi proteins of germ cells of the testis during their development, including spermatids, and those of Golgi saccular elements found in the CD of sperm transit[7,12,13], it is conceivable that protein cargos such as glycolytic proteins localize specifically to the CD by virtue of specialized Golgi protein makeup of the saccular elements adapted for future use to maintain energy production to sustain sperm motility.

A striking morphological deformation in *Sypl1* KO spermatozoa is the flagellar angulation at the site of empty CD at the neck and mid-piece area of the flagellum. This appears to occur after spermiation, as we did not find angulation in unreleased step 16 spermatids (Supplementary Fig. 2f). The homogeneous expression of HK1 evenly distributed along the entire mid-piece could result in a spontaneous burst of ATP production for sperm and this would be in the proximal epididymis, where sperm should not be motile, as they could harm each other. With HK1 restricted to the CD then ATP production would be sequentially released since the CD begins life at the neck region and then gradually migrates along the mid-piece to the annulus. By the time this occurs sperm are in the distal epididymis where the jelly-like immobilin protein immobilizes the motile sperm, preventing them from being hazardous to each other.

Another scenario as a likely secondary effect of CD malformation may be due to loss of critical proteins to regulate osmotic challenges upon sperm release from the testis. Consistent with this notion, we found the expression of PRSS21, a protein essential for sperm osmotic regulation[43], was not sequestered to the CD regions of step 16 spermatids and caput sperm (Supplementary Fig. 9). Thus, the male infertility of *Sypl1* KO mice could be caused by a combination of motility and morphological defects. We propose the CD acts as a reserve/storage site for post-testicular maturation of spermatozoa, and that SYPL1 and its partners provide an active trafficking system to concentrate critical energy and volume regulatory proteins to help sperm adapt to the changing environment before fertilization.

SYPL1 is a ubiquitously expressed physin family protein that defines constitutive transport vesicles[21]. The importance of the physin family genes in vesicle trafficking in vivo by genetic models has been unclear. Our revelation of SYPL1's crucial function in germ cell vesicular function prompts investigations of SYPL1 and related physin family proteins in other tissue contexts. For example, in the nervous system, the loss of synaptic vesicle maker protein Synaptophysin in mice does not cause a discernable vesicle phenotype[44,45], suggesting a possible compensatory role for SYPL1 or other physin proteins. Additionally, is its role in formation of a flattened saccular element not unlike those comprising Golgi ribbon. Our finding could enhance the field of Golgi investigation by adding protein players in the domain of saccular formation, especially prominent in cells following their mitosis[46]. This opens a renewed interest in studying the physin family proteins in various cellular contexts.

In conclusion, by revealing the gene function of SYPL1, we propose a molecular model of CD formation and provide the genetic insights into the molecular origin of the saccules within the forming CD during sperm maturation (Fig. 7). After the Golgi apparatus fulfills its function in acrosome formation, it emanates small SYPL1 vesicles from the TGN of late spermatids in conjunction with VAMP3 to form the saccular elements that are instrumental in focally sequestrating cargos such as HK1 to the forming CD and key to sperm fertility. Indeed, in the absence of SYPL1 and VAMP3, the vesicle operating system collapses, resulting in an "empty" CD. The exact mechanism by which this pathway operates requires the identification and characterization of more components involving thorough biochemical and genetic studies. Nevertheless, our data strongly support the hypothesis that the CD is a functional entity rather than a redundant remnant of cytoplasm. In fact, the enrichment of SYPL1 and VAMP3 is conserved, also appearing in the CDs of human spermatozoa (Supplementary Fig. 10). Our findings provide a starting point to further pinpoint the functional importance of CDs in mammalian spermatozoa.

## Methods

### Ethics statement

All mice were bred and housed under specific pathogen-free conditions with controlled temperature (20–25 °C), 50–70% humidity and exposed to a constant 12-h light–dark cycle in the animal facilities of Michigan State University. All the animal procedures were reviewed and approved by the Institutional Animal Care and Use Committee of Michigan State University. All experiments with mice were performed ethically according to the Guide for the Care and Use of Laboratory Animals and institutional guidelines.

### Mice

Mice with exon 3 of the *Sypl1* gene deleted were obtained from the International Mouse Phenotyping Consortium (IMPC). The *Sypl1* allele (C57BL/6N-Sypl<tm1b(KOMP)Wtsi > /Tcp) was backcrossed to C57BL/6 N background for six generations. For *Sypl1* mice genotyping, mouse tail snips were digested in 80 µl lysis buffer (50 mM NaOH) for 30 min at 98 °C and neutralized by 20 µl 1 M Tris-HCl pH 7.6. Digested tail DNA was used for PCR using 2× AccuStart PCR master mix (Quanta Biosciences). PCR reactions were carried out with the following amplification conditions: 35 cycles of 95 °C for 30 s, 60 °C for 30 s, 72 °C for 30 s on C1000 Touch Thermal Cycler (Bio-Rad). Primers used were: *Sypl1*-P1: CCATTACCAGTTGGTCTGGTGTC; *Sypl1*-P2: GTAGGAGAAG GACATTACGTGCAGGC; *Sypl1*-P3: GATGTGCATGTTTTATTTGTGGTC. WT allele yields a 288 bp band, and mutant allele yields a 599 bp band. Only male mice were used in the current study.

### Histology

Mouse testes and epididymides were dissected and fixed in Bouin's solution (HT10132, Sigma-Aldrich, USA) overnight at 4 °C on a rotator, then the testes were cut into two pieces, after two washes by 0.01 M PBS (pH 7.4), and dehydrated in gradient series of different concentrations ethanol, followed by removing the ethanol by xylene twice, embedded in paraffin, and 5 µm sections were cut. For the histological analysis of testes and epididymides, sections were stained with hematoxylin and eosin after dewaxing and rehydration.

### RT-PCR

Total RNA was extracted from mouse tissues using TRIzol Reagent (15596026, ThermoFisher). A total of 1 µg of total RNA was treated with DNase I (M0303S, New England Biolabs) and then was reverse transcribed using iScript cDNA Synthesis Kit (1708890, Bio-Rad) according to manufacturer's instructions. cDNA was diluted 5-fold and PCR was conducted using AccuStart II GelTrack PCR SuperMix (95136-04 K, Quantabio) in C1000 Touch Thermal Cycler (1851148, Bio-Rad). Thermal cycling condition is 94 °C 15 s, 60 °C 15 s and 72 °C 30 s. Primers used for PCR are: *Sypl1* forward 5′- CTCGAGTGGTTTGCCTCCAT-3′, *Sypl1* reverse 5′-GTCGATCATGGGCAGTTTGC-3′, *Vamp3* forward 5′-AGGGATCAGTGTCCTGGTGA-3′, *Vamp3* reverse 5′-TGCCTTGAACAAA ACCCCCT-3′, *Actb* forward 5′-CATCCGTAAAGACCTCTATGCCAAC-3′, *Actb* reverse 5′-ATGGAGCCACCGATCCACA-3′.

### Immunofluorescence (IF) staining for testes and epididymides frozen sections

Mouse testes and epididymides were fixed in 4% paraformaldehyde (PFA) overnight at 4 °C, after washed in 0.01 M PBS (pH 7.4), the testes were perfused in 30% sucrose, and embedded in Tissue-Tek O.C.T Compound (4583, Sakura Finetek, Torrance, CA). Tissue sections were cut at 7.0 µm and mounted on silanized slides. Then the sections were fixed with 10% buffered formalin phosphate (SF100-4, Fisher Scientific, Fair Lawn, NJ) for 10 min at room temperature (RT). After three washes for 5 min with 0.01 M PBS, nonspecific binding was blocked by incubating the sections in 5% normal goat serum (NGS) for 30 min at RT. The sections were then incubated with rabbit anti-SYPL1 (1:50, Abcam, ab184176), rabbit anti-VAMP3 (1:100, ab43080, Abcam), rabbit anti-ACRV1 (1:100, 14040-1-AP, Proteintech), rabbit anti-hexokinase 1 (HK1, 1:400, ab3543, Millipore), rabbit anti-PGK2 (1:400, ab183031, Abcam), goat anti-PRSS21 (1:500, PA5-47879, ThermoFisher) in 5% NGS at 37 °C for 2 h. After PBS washed, they were incubated with secondary antibody (1:500, Alexa-Fluor 488 or 555, Life Technologies) for 1 h at RT.

Finally, the sections were mounted using Vectorshield mounting media with DAPI (H1200, Vector Laboratory) after 3 washes with 0.01 M PBS (pH 7.4). Fluorescence was observed and photographed with a confocal microscope (FluoView1000, Olympus, Japan). Co-staining of organelle markers, HK1, ACRV1, VAMP proteins and SYPL1, HK1 and VAMP3 were performed according to the data sheet of Zenon® Antibody Labeling Kits (Z25302, Z25305, ThermoFisher). Briefly, all the sections were washed in 0.01 M PBS for 5 min × 3 times, followed by washing in PBS containing 0.02% Triton X 100-PBS (PBST), and nonspecific binding was blocked by incubating the sections in 5% NGS for 30 min at RT. Then the sections were incubated with a mixture of Alexa Fluor 555 conjugated rabbit anti-SYPL1 and Alexa Fluor 488 conjugated rabbit anti-VAMP3 (or VAMP2, VAMP4, HK-I, ACRV1, AIF, EEA1, PDI, RCAS1, GM130, and Golgin97) for 1.5 h at RT. After washing in PBST for 10 min × 3 times and in PBS for 5 min × 2 times, the sections were mounted as above. Fluorescence was observed and photographed with a confocal microscope (FluoView1000, Olympus, Japan).

## Western blot analysis

Mouse tissues were homogenized and epididymal sperm were sonicated in RIPA buffer with protease inhibitor (A32963, Pierce) and 1 mM PMSF. Tissue protein lysates were separated by 4–20% SDS-PAGE gel and transferred to polyvinlylidenedifluoride (PVDF) membranes (Bio-Rad). The membranes were blocked in 5% non-fat milk, subsequently incubated with primary antibodies in blocking solution overnight at 4 °C. After washed in TBST, the membranes were incubated with HRP conjugated goat anti-rabbit IgG (1:5000, 1706515, Bio-Rad) or HRP conjugated rabbit anti-goat IgG (1:5000, 1721034, Bio-Rad) for 1 h. Membranes were incubated with chemiluminescent substrates (Bio-Rad) for detection in a molecular Imager ChemiDoc XRS+ imaging system (Bio-Rad) after 3 washes in TBST. β-Actin was visualized by incubating with anti-β-Actin-HRP (1:10000, A3854, Sigma-Aldrich) after stripping and washing. The primary antibodies used are as follows: rabbit anti-SYPL1 (1:2000, Abcam, ab184176), rabbit anti-VAMP2 (1:10000, ab181869, Abcam), rabbit anti-VAMP3 (1:10000, ab43080, Abcam), rabbit anti-VAMP4 (1:5000, 10738-1-AP, Proteintech), rabbit anti-AIF (1:1000, 5318, Cell Signaling Technology), rabbit anti-HK1 (1:4000, ab3543, Millipore), rabbit anti-PGK2 (1:5000, ab183031, Abcam), goat anti-PRSS21 (1:500, PA5-47879, ThermoFisher).

## Sperm count

The cauda epididymis was dissected from adult HET and KO mice. Sperm were allowed to exude from the incisions of cauda epididymis in 2 ml PBS for 15 min in a 37 °C incubator. Then pipette 0.5 ml of the suspending solution of incubated sperm to 2 ml 0.01 M PBS, incubate the mixture at 65 °C for 10 min, and transfer 10 μl to the hemocytometer for sperm counting under a light microscope.

## Computer-assisted sperm analysis (CASA)

Sperm were isolated by shredding cauda epididymis in 2 mL HTF media (2002, InVitroCare). After incubating at 37 °C for 20 min, sperm were diluted 3-fold with HTF media. CASA was done using IVOS II (Hamilton Thorne, USA). At least 600 sperm from each sample were examined. Motion parameters of motility (%), progressive motility (%), average path velocity (VAP), straight-line velocity (VSL) and curvilinear velocity (VCL) were collected.

## Preparation of testes and epididymal sperm smears

Testes spermatozoa isolation was performed as described previously[47]. Briefly, WT or HET and KO testes were dissected from the mice bodies, and the albuginea was removed, then the darkest regions of seminiferous tubules were cut under a stereomicroscope and put on a slide in 200 μl 0.01 M PBS. Subsequently, the stage 16 spermatozoa were expelled using a coverslip, 5 μl of the suspending solution

containing the spermatozoa was pipetted and spread on a new slide after removing the coverslip, the smear slides were dried at 37 °C, and then IF or hematoxylin staining was performed. Epididymal sperm smear preparation was performed as reported (Yuan et al., 2013). Briefly, spermatozoa from mice caput or cauda epididymis were smeared onto Superfrost Plus slides (Fisher Scientific, Hampton, NH) and dried in an oven at 37 °C. Human sperm samples were kept in a 5% $CO_2$ incubator at 37 °C for 30 min to allow liquefaction. Then the liquefied semen samples were fractionated by the swim-up method as described following: Liquefied (0.2 ml in culture tube) semen sample were lay gently under 2 ml of 0.01 M PBS and then incubated for 30 min at 37 °C in a $CO_2$ incubator. The supernatant containing sperm was smeared onto Superfrost Plus slides and dried as above. Meanwhile, sperm were collected by centrifugation at 300 × g for 5 min at 4 °C. The pellet was utilized for protein sample preparations.

## Sperm morphology

Adult HET and KO sperm smears were stained with hematoxylin, dehydrated, mounted, observed and photographed using EVOS FLc microscope (Life Technologies). Adult HET and KO caput or cauda epididymal sperm were isolated, loaded into 80μm-depth 2X-CEL chambers (Hamilton Thorne Research, Beverly, MA, USA) and phase contrast photos were photographed using EVOS FLc microscope (Life Technologies).

## IF of testes and epididymal sperm smears

The sperm smear was fixed with 10% formalin for 10 min at RT. After rinsed in 0.01 M PBS, the smears were incubated with 0.5% Triton X 100-PBS for 5 min, then non-specific binding sites were blocked with 5% NGS in 0.2% Triton X 100-PBS for 30 min at RT. Subsequent steps for immunofluorescence of the smears were performed as described above. The primary antibodies used are as follows: rabbit anti-SYPL1 (1:50, Abcam, ab184176), rabbit anti-VAMP3 (1:100, ab43080, Abcam), rabbit anti-HK1 (1:100, ab3543, Millipore), rabbit anti-PGK2 (1:100, ab183031, Abcam), goat anti-PRSS21 (1:50, PA5-47879, ThermoFisher). After washing in PBST for 10 min × 3 times and in PBS for 5 min × 2 times, the sections were incubated with fluorescent secondary antibodies, and then mounted as above. Fluorescence was observed and photographed using EVOS FLc microscope (Life Technologies).

## Fertility tests

In order to examine the fertility of *Sypl1* KO mice, adult male mice of HET (n = 3) or KO (n = 10) were partnered with adult WT female C57BL/6 N mice for 5 months. At the same time, adult female mice of KO were mated with HET male mice. Every day during the cohabitation, females were examined for vaginal plugs as evidence of mating. Plugged mice that gave birth to offspring were recognized as pregnant. The number of total litters and offspring were recorded.

## Scanning and transmission electronic microscopy (SEM and TEM)

For SEM, testicular and epididymal sperm were isolated and fixed in 4% glutaraldehyde; then SEM followed standard protocols performed at the Laboratory of Electron Microscopy, Michigan State University Center for Advanced Microscopy. Images were taken with JEOL 7500 F ultra-high-resolution scanning electron microscope (Japan Electron Optics Laboratory, Japan). For TEM, the testes and epididymides were fixed in 2.5% glutaraldehyde and 2% PFA. After primary fixation, samples were washed with 0.1 M cacodylate buffer and postfixed with 1% osmium tetroxide in 0.1 M cacodylate buffer, dehydrated in a gradient series of acetone and infiltrated and embedded in Spurr. A total of 70 nm thin sections were obtained with a Power Tome Ultramicrotome (RMC, Boeckeler Instruments. Tucson, AZ) and post-stained with uranyl acetate and lead citrate. Images were taken with JEOL 100CX

Transmission Electron Microscope (Japan Electron Optics Laboratory, Japan) at an accelerating voltage of 100 kV.

## Immuno-electron microscopy (IEM)

IEM was performed according to the previous report[48] with some modifications. Mouse testes and epididymides were fixed in 4% PFA overnight at 4 °C, after washed in 0.01 M PBS (pH 7.4), the testes were perfused in 30% sucrose, and embedded in O.C.T reagent. Tissue sections were cut at 10μm and mounted on Superfrost Plus microscope slides (12-550-15, Fisher Scientific). The sections were washed in PBS for 5 min, then in PBS with 0.05 M glycine for 10 min, and nonspecific binding was blocked by incubating the sections in 5% normal goat serum (NGS) for 30 min at RT. The sections were then incubated with rabbit anti-SYPL1 (1:50, Abcam, ab184176) in 5% NGS at 4 °C for overnight. After PBS washed, they were incubated with Nanogold-Fab' Goat anti-Rabbit IgG conjugated with 1.4-nm gold particles (2004, Nanoprobes) for 1 h at RT. After PBS washed, they were fixed in 2% glutaraldehyde for 10 min. Following silver enhancement by HQ Silver kit (2012, Nanoprobes) according to manufacturer's manual, the sections were washed in 3% sodium thiosulfate for 3 min, postfixed in 1% osmium tetroxide for 10 min, dehydrated in a gradient series of ethanol, infiltrated and embedded in Spurr (14300, Electron Microscopy Sciences). Around 70 nm thin sections were obtained with a Power Tome Ultramicrotome (RMC, Boeckeler Instruments. Tucson, AZ) and post-stained with uranyl acetate and lead citrate. Images were taken with JEOL 1400 Flash Transmission Electron Microscope (Japan Electron Optics Laboratory, Japan).

## Co-immunoprecipitation (Co-IP)

WT testes were homogenized at 18000 rpm for 30 s and lysed with 1% NP40 buffer supplemented with protease inhibitor and 1 mM PMSF for 40 min on ice. The lysate was centrifuged at $12000 \times g$ for 30 min and the supernatant was filtered through 0.22 μm filter, followed by incubating with 100 μl protein-A-agarose beads (11134515001, Roche, Germany) for 3 h at 4 °C. Precleared lysate was incubated with anti-SYPL1 rabbit monoclonal antibody and anti-VAMP3 rabbit polyclonal antibody overnight at 4 °C, respectively. A total of 50 μl protein-A-agarose beads were added to each lysate and incubated for 3 h at 4 °C, then the beads were washed three times with 1% NP40 buffer and heated for 10 min at 100 °C in SDS loading buffer. And the samples were subjected to Western blot as above.

## CD-bearing sperm and saccular elements-bearing sperm counting

Epididymal CD-bearing sperm were counted under the phase contrast microscope from at least 3 pairs of adult HET and KO mice. Testicular and epididymal saccular elements-bearing sperm were counted in TEM photos from 3 pairs of adult HET and KO mice.

## Statistics and reproducibility

All data are mean ± SEM and all statistical analyses between groups were analyzed by two-sided Student's $t$-test, a value of $p < 0.05$ was considered to be statistically significant (*$p < 0.05$; **$p < 0.01$; ***$p < 0.001$; ****$p < 0.0001$). Each in vivo experiment was repeated independently for at least three times and achieved similar results.

## Reporting summary

Further information on research design is available in the Nature Portfolio Reporting Summary linked to this article.

## Data availability

All data supporting the findings of this study are available within the article and its Supplementary Information file or Source Data file. Source data are provided as a Source Data file. Source data are provided with this paper.

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

## Acknowledgements

We thank X. Cheng for critical reading of the manuscript, L. Nutter and the Center for Phenogenomics for targeted *Sypl1* mice (C57BL/6N-Sypl<tm1b(KOMP)Wtsi > /Tcp). Jiali Liu was supported by the National Natural Science Foundation of China [No. 32171111 to J.L.] and the Beijing Natural Science Foundation [No. 5222015 to J.L.], Chen Chen was supported in part by MSU AgBioResearch funds and NIH grants R01HD084494 and R01GM132490.

## Author contributions

J.L. and C.C. designed research. J.L., Y.W., A.F.M., and A.W. performed histology, immunostaining, Western blot, SEM and TEM. D.D. and J.M.M. performed fertility test and contributed to mouse breeding. C.W. and X.Y. performed IEM. J.C., R.A.H., and L.H. analyzed the histological defects of *Sypl1* KO mice. J.L., L.H., R.A.H., and C.C. wrote the manuscript. C.C. supervised the project.

## Competing interests
The authors declare no competing interests.
