## [Peer Review File · Nature Communications]

SYPL1 defines a vesicular pathway essential for sperm cytoplasmic droplet formation and male fertilityREVIEWER COMMENTS

Reviewer #1 (Remarks to the Author):

In this manuscript, the authors provide solid evidence for the role of SYPL1 (synaptophysin-like protein 1) in formation of the cytoplasmic droplet (CD) and its enclosed metabolic enzymes in mouse spermatids. Through detailed and elegant microscopy they reveal a novel pathway of vesicle traffic and implicate roles not only for SYPL1 but also for VAMP3 (vesicle-associated membrane protein 3, a v-SNARE protein).

The study was developed to resolve the problem of how hexokinase 1 gets to and is enriched in the CD, the underlying hypothesis being that membranous elements were key. Because of its high expression level in male germ cells, SYPL1 was selected for mutational analysis. The abnormal sperm and the infertility of mutant males provided justification for the detailed studies reported in this manuscript. The infertility phenotype analyses are appropriately conducted, and the morphological analyses are meticulous and outstanding in quality. Detailed light and electron microscopy revealed origins of SYPL1 and the CD in the Golgi network, and here, the schematic cartoons (e.g., Fig. 4b) are helpful to the reader lacking familiarity with the differentiation of these structures in spermatids.

Interestingly, in addition to SYPL1, a member of the VAMP protein family, VAMP3, is also involved. The investigators first discovered this by a good guess – VAMP proteins associate with the synaptophysin family proteins – and validated it by expression and immunolocalization and immunoprecipitation, which showed interaction of SYPL1 and VAMP3. Analysis of the *Sypl1* KO provided some evidence that SYPL1 may control expression of VAMP3. However, this was not validated with RNA analysis (to determine if effect is transcriptional or post-transcriptional) or quantitative protein analyses (such as western blotting), which would be important evidence bolstering the conclusion. Further morphological analyses (Fig. 6) suggest improper saccule formation in *Sypl1* KO germ cells, perhaps due to improper VAMP3 function. This raises a conundrum: why doesn't the *Vamp3* KO exhibit male infertility, perhaps even phenocopying the *Sypl1* KO? (Yang et al. 2001. *Mol Cell Biol* 21:1573, <https://www.informatics.jax.org/marker/MGI:1321389>). Double mutant analysis would help resolve this problem. At the very least, the authors should address this issue and the implications of other interpretations for their model.

Overall, the data in this report are quite strong and visually informative. The interpretations, for the most part, are convincing and illustrated with good diagrams. The idea that the CD saccules contribute to sperm motility is especially interesting and has implications for contraceptive interference. One issue of concern is uncertainty about the role of VAMP3 – is it required and what does it do? This is inadequately addressed, and appropriate analyses to demonstrate that *Vamp3* is “genetically downstream” of *Sypl1* have not been done. Nonetheless, the elegant morphological results will surely be appreciated by reproductive biologists, especially those interested in the extensive cytodifferentiation of sperm cells. However, it is not clear if the larger audience of CMLS readers will find this more than of “niche” interest. The authors could broaden the general appeal by addressing the relevance for vesicle pathways in developmental processes (e.g., neurogenesis and neuronal function references cited in Line 318).

Minor editorial comments:

Line 47: delete “become haploid” (inaccurate because spermatids are already haploid, but mentioned correctly in line above)

Line 150: “along” does not make sense; perhaps “although” is intended?

Line 229: this sentence does not seem complete; suggest adding “to define” at the end

Line 299: perhaps this should be a “novel cellular pathway” given that the analyses are almost entirely morphological and the genetic relationships between Sypl1 and Vamp 3 (dominance, epistasis) have not been resolved

Line 316: the use of “genetically downstream” is problematical, as the appropriate analyses to demonstrate this have not been performed

Line 382: it is not clear what “conservatively” means here – perhaps the authors mean “the enrichment of SYPL1 and VAMP3 is conserved, also appearing in the CDs of human spermatozoa”

Reviewer #2 (Remarks to the Author):

This study identifies SYPL1 as a novel factor in accumulation of saccular elements in the cytoplasmic droplet (CD) in sperm. CDs in sperm have long been noted but their formation and function remain enigmatic. This study identifies an uncharacterized membrane protein SYPL1 as a component of CD. SYPL1 is mainly expressed in testis. Inactivation of SYPL1 did not affect sperm production, but the KO sperm were immotile and had abnormal head foldback or flagellar foldback defects. EM showed absence of saccular elements in CDs of Sypl1 KO sperm, implying a critical role for SYPL1 in this process. They found that glycolytic enzymes HK1 and PGK2 normally accumulated in CDs of wild type sperm were absent in CDs of Sypl1 KO sperm. SYPL1 is associated with VAMP3, a membrane-associated protein involved in vesicle trafficking. SYPL1 colocalizes with VAMP3 in the trans-Golgi network. This study is very comprehensive. The data are of high quality and support the conclusions. The manuscript is well written and prepared. SYPL1 would be the first functional component of CDs. This study provides interesting insights into the formation of CDs. The findings are very significant for the reproduction field.

Minor concerns:

1) Mating test results (Table 1): The mating test result for KO males was not clear. 10 KO males were tested. 2 KO males did produce two litters. It is not clear whether each KO male gave two litters or each KO gave one litter only. I assumed that the other 8 KO males never produced a litter, if not so, please state it in Results section. It can be stated that 8 out of 10 KO males tested did not give rise to litters and that the remaining two KO males did produce a litter.

2) Fig. 3a-d: the colors of arrows are a bit confusing, albeit denoted in the legend. It is better to use the same colors. In Fig. 3b, the arrow colors for CD and RB should be changed to yellow and red respectively, to match their colors in 3a. In Fig. 3d, the arrow color for RB should be red.

3) Lines 225-229: This paragraph “TEM analysis of the saccules in the wildtype CD revealed that while some...” does not seem to belong to the Results section. No figure was cited in this paragraph. I think that it is better to be moved to or incorporated into the Discussion.

Reviewer #3 (Remarks to the Author):

Liu et al. established Sypl1 KO mice and carefully described the phenotype using immunohistochemistry, transmission electron microscopy and immunoblots. Marker proteins at each stage of spermiogenesis were used to define the origin of SYPL1 and determine its role in cytoplasmic droplet (CD) formation. Additional experiments on the function of SYPL1 in CD formation and its effect on sperm motility would be informative.

Major concerns:

1. The head/midpiece of Sypl1 KO sperm are angulated, but they still have ~10% normal motility. Mutant sperm have normal acrosome, midpiece, but fail to accumulate glycolytic enzymes in cytoplasmic droplets (CDs). The authors should compare KO and control sperm with WT eggs for in vitro fertility with and without cumulus cells as well as with and without zonae pellucidae.
2. The authors should report number of viable KO sperm as well as the percent of rapid motility, progressive motility, VSL, VCL, etc. (CASA).
3. Sypl1 KO sperm abnormalities becomes more severe following epididymal transit. During maturation, sperm interact with epididymal vesicles (epididymosome). Therefore, the authors should compare the proteome from the four segments (initial segment, caput, corpus, caudal) of the epididymis in KO and WT. This should provide molecular insight into mechanisms associated with the phenotype.
4. Reduce levels of PRSS21 (Testisin) was observed in KO mice and the authors should clarify when/where loss of protein occurred during spermatogenesis.
5. The authors should determine the phosphorylation status in WT and KO sperm under capacitating and non-capacitating conditions.

Minor concerns:

1. Fig. 1d: PRSS21 (36.2 kD) is most abundant in testis; the other faint bands are probably non-specific binding. Is there a RT-PCR signal in tissues other than testis?
2. Fig. 1f: Please clarify the sperm origin (testis vs. epididymis) and provide a brightfield image for localization.
3. Fig. 2f-g, S2f, S3: Please include scale bars.

REVISION SUMMARY

We thank all the reviewers for their insightful comments and suggestions. In our revised manuscript, we have addressed all the questions raised by reviewers by adding new data and text revisions.

In the revised manuscript, we have added the following new figures and tables:

- Fig. 1f: a bright field image was added.
- Fig. S2b and S2c: new sperm motility and velocity data by CASA were added.
- Fig. S8a: new RT-PCR results of *Syp11* and *Vamp3* mRNA in HET and KO testes were added.
- Fig. S9a: PRSS21 immunostaining in HET and KO testes was added.
- Table S1: fertility test details were added.

We have highlighted all the changes in yellow in manuscript text and below is our point-by-point response to the reviewers' comments in blue.

RESPONSE TO REFEREES

Reviewer #1 (Remarks to the Author):

In this manuscript, the authors provide solid evidence for the role of SYPL1 (synaptophysin-like protein 1) in formation of the cytoplasmic droplet (CD) and its enclosed metabolic enzymes in mouse spermatids. Through detailed and elegant microscopy they reveal a novel pathway of vesicle traffic and implicate roles not only for SYPL1 but also for VAMP3 (vesicle-associated membrane protein 3, a v-SNARE protein).

The study was developed to resolve the problem of how hexokinase 1 gets to and is enriched in the CD, the underlying hypothesis being that membranous elements were key. Because of its high expression level in male germ cells, SYPL1 was selected for mutational analysis. The abnormal sperm and the infertility of mutant males provided justification for the detailed studies reported in this manuscript. The infertility phenotype analyses are appropriately conducted, and the morphological analyses are meticulous and outstanding in quality. Detailed light and electron microscopy revealed origins of SYPL1 and the CD in the Golgi network, and here, the schematic cartoons (e.g., Fig. 4b) are helpful to the reader lacking familiarity with the differentiation of these structures in spermatids.

We thank the Reviewer for the elegant summary of the quality and significance of this manuscript.

Interestingly, in addition to SYPL1, a member of the VAMP protein family, VAMP3, is also involved. The investigators first discovered this by a good guess – VAMP proteins associate with the synaptophysin family proteins – and validated it by expression and immunolocalization and immunoprecipitation, which showed interaction of SYPL1 and VAMP3. Analysis of the *Sypl1* KO provided some evidence that SYPL1 may control expression of VAMP3. However, this was not validated with RNA analysis (to determine if effect is transcriptional or post-transcriptional) or quantitative protein analyses (such as western blotting), which would be important evidence bolstering the conclusion.

Response: We thank the Reviewer for these insightful comments and suggestions. Following suggestions, we performed RNA analysis (RT-PCR) and show that the mRNA level of *Vamp3* in the KO testes was equivalent to that of HET (**see below Fig. R1-1**), suggesting that the regulation of VAMP3 by SYPL1 is post-transcriptional. Western blotting shows diminished VAMP3 in KO caput and cauda sperm although testicular VAMP3 level is not significantly reduced due to high VAMP3 expression in spermatocytes (**Fig. S8**). This is consistent with what we showed in **Fig. 5d-f** that VAMP3 was lost in elongated spermatids in situ. Collectively, these data indicate that VAMP3 protein is affected by SYPL1 deficiency, which is likely due to the loss of SYPL1-VAMP3 interaction. Related changes have been added to the revised version of the manuscript highlighted by yellow.

Fig. R1-1. RT-PCR of *Sypl1* and *Vamp3* mRNA in HET and KO testes. β -Actin served as an internal control.

Further morphological analyses (Fig. 6) suggest improper saccule formation in *Sypl1* KO germ cells, perhaps due to improper VAMP3 function. This raises a conundrum: why doesn't the *Vamp3* KO exhibit male infertility, perhaps even phenocopying the *Sypl1* KO? (Yang et al. 2001. Mol Cell Biol 21:1573, <https://www.informatics.jax.org/marker/MGI:1321389>). Double mutant analysis would help resolve this problem. At the very least, the authors should address this issue and the implications of other interpretations for their model.

Response: Indeed, *Vamp3* KO males are reported to be fertile (Borisovska et al., 2005; Schraw et al., 2003). One logical explanation for the fertility differences between *Sypl1* KO and *Vamp3* KO is that the molecular functions of these two proteins are expected

to be different. As a v-SNARE, VAMP3 is known for its function in vesicle fusion in various tissue contexts (Banerjee et al., 2017; Gordon et al., 2017; Hu et al., 2007; Nozawa et al., 2017; Yang et al., 2001). We do observe the vesicle fusion phenomenon in wild-type spermatids during CD formation which could be mediated by VAMP3. Disrupting vesicle fusion by VAMP3 deficiency may not affect saccule sequestration to the CD and thus producing CD with sufficient saccular contents for proper sperm motility and function. This could be the reason why *Vamp3* KO mice are fertile. By contrast, in addition to regulating VAMP3, SYPL1 may interact with additional proteins to regulate both vesicle formation and transport (Haass et al., 1996). Indeed, in **Fig. S7d**, we show that SYPL1 expression preceded VAMP3 in the TGN region of the Golgi apparatus in WT spermatids, further implicating its role in vesicle initial formation and therefore functionally distinct from VAMP3. We agree with the reviewer that to understand the precise function of VAMP3 and SYPL1 requires detailed analysis of single KOs and the double KO, which would be suitable for future studies. We have added a short discussion about the explanation of fertility differences in the discussion section of the revised manuscript.

Overall, the data in this report are quite strong and visually informative. The interpretations, for the most part, are convincing and illustrated with good diagrams. The idea that the CD saccules contribute to sperm motility is especially interesting and has implications for contraceptive interference. One issue of concern is uncertainty about the role of VAMP3 – is it required and what does it do? This is inadequately addressed, and appropriate analyses to demonstrate that *Vamp3* is “genetically downstream” of *Sypl1* have not been done.

Response: We have explained the potential role of VAMP3 above. We acknowledge that the description of “genetically downstream” is not appropriate and therefore have deleted this statement in the revised manuscript.

Nonetheless, the elegant morphological results will surely be appreciated by reproductive biologists, especially those interested in the extensive cytodifferentiation of sperm cells. However, it is not clear if the larger audience of CMLS readers will find this more than of “niche” interest. The authors could broaden the general appeal by addressing the relevance for vesicle pathways in developmental processes (e.g., neurogenesis and neuronal function references cited in Line 318).

Response: We thank the reviewer for this important suggestion. SYPL1 is a ubiquitously expressed physin family protein that defines constitutive vesicles (Haass et al., 1996). This is the first study to report the in vivo function of SYPL1 in mice. SYPL1 is also expressed in the brain and other tissues. While loss of SYPL1 paralog synaptic vesicle marker Synaptophysin in mice does not cause an obvious phenotype in the nervous system (Eshkind and Leube, 1995; McMahon et al., 1996), suggesting

a possible compensatory role for SYPL1 or other physin family proteins in the nervous system. Additionally, is its role in formation of a flattened saccular element not unlike those comprising Golgi ribbon. Our finding could enhance the field of Golgi investigation by adding new protein players in the domain of saccular formation, especially prominent in cells following their mitosis (Pfeffer, 2013). We have added a new paragraph in the Discussion section of the revised version to expand broader significance. We believe it will be of interest to CMLS biologists studying intracellular trafficking, membrane protein and Golgi biology, gene expression and function, and genetics in various tissue contexts.

Minor editorial comments:

Line 47: delete “become haploid” (inaccurate because spermatids are already haploid, but mentioned correctly in line above)

Response: We have deleted “become haploid”.

Line 150: “along” does not make sense; perhaps “although” is intended?

Response: Changed to “although”.

Line 229: this sentence does not seem complete; suggest adding “to define” at the end

Response: We have modified as suggested.

Line 299: perhaps this should be a “novel cellular pathway” given that the analyses are almost entirely morphological and the genetic relationships between Syp11 and Vamp 3 (dominance, epistasis) have not been resolved

Response: We have changed it to a “novel cellular pathway”.

Line 316: the use of “genetically downstream” is problematical, as the appropriate analyses to demonstrate this have not been performed

Response: We have removed “genetically downstream”.

Line 382: it is not clear what “conservatively” means here – perhaps the authors mean “the enrichment of SYPL1 and VAMP3 is conserved, also appearing in the CDs of human spermatozoa”

Response: We thank the reviewer for this suggestion and have changed the statement accordingly.

Reviewer #2 (Remarks to the Author):

This study identifies SYPL1 as a novel factor in accumulation of saccular elements in the cytoplasmic droplet (CD) in sperm. CDs in sperm have long been noted but their formation and function remain enigmatic. This study identifies an uncharacterized membrane protein SYPL1 as a component of CD. SYPL1 is mainly expressed in testis. Inactivation of SYPL1 did not affect sperm production, but the KO sperm were immotile and had abnormal head foldback or flagellar foldback defects. EM showed absence of saccular elements in CDs of Sypl1 KO sperm, implying a critical role for SYPL1 in this process. They found that glycolytic enzymes HK1 and PGK2 normally accumulated in CDs of wild type sperm were absent in CDs of Sypl1 KO sperm. SYPL1 is associated with VAMP3, a membrane-associated protein involved in vesicle trafficking. SYPL1 colocalizes with VAMP3 in the trans-Golgi network. This study is very comprehensive. The data are of high quality and support the conclusions. The manuscript is well written and prepared. SYPL1 would be the first functional component of CDs. This study provides interesting insights into the formation of CDs. The findings are very significant for the reproduction field.

We thank the reviewer for concisely summarizing all key discoveries and significance of this work.

Minor concerns:

1) Mating test results (Table 1): The mating test result for KO males was not clear. 10 KO males were tested. 2 KO males did produce two litters. It is not clear whether each KO male gave two litters or each KO gave one litter only. I assumed that the other 8 KO males never produced a litter, if not so, please state it in Results section. It can be stated that 8 out of 10 KO males tested did not give rise to litters and that the remaining two KO males did produce a litter.

Response: We thank the reviewer for pointing this out and the reviewer was correct. 8 KO males never produced a litter. 2 KO males produced only 1 litter each over a 5-month breeding period. We have modified **Table S1** and the sentences in the text as the reviewer suggested.

2) Fig. 3a-d: the colors of arrows are a bit confusing, albeit denoted in the legend. It is better to use the same colors. In Fig. 3b, the arrow colors for CD and RB should be changed to yellow and red respectively, to match their colors in 3a. In Fig. 3d, the arrow color for RB should be red.

Response: We have revised the arrow colors in **Fig. 3b and 3d** in the revised manuscript as suggested.

3) Lines 225-229: This paragraph “TEM analysis of the saccules in the wildtype CD revealed that while some...” does not seem to belong to the Results section. No figure was cited in this paragraph. I think that it is better to be moved to or incorporated into the Discussion.

Response: We thank the reviewer for this helpful suggestion. We have removed this paragraph to make the results section more coherent.

Reviewer #3 (Remarks to the Author):

Liu et al. established Sypl1 KO mice and carefully described the phenotype using immunohistochemistry, transmission electron microscopy and immunoblots. Marker proteins at each stage of spermiogenesis were used to define the origin of SYPL1 and determine its role in cytoplasmic droplet (CD) formation. Additional experiments on the function of SYPL1 in CD formation and its effect on sperm motility would be informative.

Major concerns:

1. The head/midpiece of Sypl1 KO sperm are angulated, but they still have ~10% normal motility. Mutant sperm have normal acrosome, midpiece, but fail to accumulate glycolytic enzymes in cytoplasmic droplets (CDs). The authors should compare KO and control sperm with WT eggs for in vitro fertility with and without cumulus cells as well as with and without zonae pellucidae.

Response: We thank the reviewer for this suggestion. Our fertility test experiments demonstrated that 2 KO males out of 10 did produce 1 litter of pups each (see **Table S1** below), indicating that although the KO sperm motility is significantly reduced, they still can naturally fertilize the egg. We hope this will answer the reviewer’s question on KO sperm’s fertility potential.

Table S1 Fertility tests within 5 months. “**” represents significant difference.

	♂HET (N=3)	♂HET (N=4)	♂HET (N=4)	♂KO (N=10)
	X	X	X	X
	♀WT	♀HET	♀KO	♀WT
Litter size (pups/litter)	6.51±0.18	6.43±0.28	6.29±0.45	≈ 0* 8 male KO: no pups 1 male KO: 1 litter of 2 pups 1 male KO: 1 litter of 5 pups
Litter interval (Litter/month)	1.13±0.06	1.15±0.05	1.15±0.05	≈ 0*

2. The authors should report number of viable KO sperm as well as the percent of rapid motility, progressive motility, VSL, VCL, etc. (CASA).

Response: We thank the reviewer for this important suggestion. We have performed the CASA analysis and added new results as follows (**Fig. R3-1**), which have been included in **Fig. S2** of the revised version.

Fig. R3-1. Sperm motility analyses by CASA. **a.** The proportion of motile and progressive spermatozoa in the cauda epididymis of HET and KO mice. **b** Evaluation of sperm motility velocity. VAP: average path velocity, VSL: straight-line velocity, VCL: curvilinear velocity. All data are presented as the mean ± SEM (***, $P < 0.001$; ****, $P < 0.0001$).

3. Sypl1 KO sperm abnormalities becomes more severe following epididymal transit. During maturation, sperm interact with epididymal vesicles (epididymosome). Therefore, the authors should compare the proteome from the four segments (initial segment, caput, corpus, caudal) of the epididymis in KO and WT. This should provide molecular insight into mechanisms associated with the phenotype.

Response: Because there is a complex protein composition within the epithelial cells of the epididymis, we cannot be sure that if the data do show differences in the proteome from different segments that it is due to epididymosomes, which would not be able to further modify sperm abnormalities, the proteins they carry should not be those that affect sperm morphology, they deliver RNA (Sharma et al., 2018) and other non-structural proteins (Barrachina et al., 2022). Our study focuses on the events of the final process of spermiogenesis in the testis, and for the proteomic analysis of different segments of the epididymis, a large amount of data is available, it may be difficult to assign, and in the case of our study, these differences may be secondary effects of sperm release into the epididymis in the testis after SYPL1 deletion, rather than a direct effect. One hypothesis to test in the future would be that loss of SYPL1 in sperm could indirectly affect the sperm ability to interact with epididymosomes.

4. Reduce levels of PRSS21 (Testisin) was observed in KO mice and the authors should clarify when/where loss of protein occurred during spermatogenesis.

Response: We thank the reviewer for this suggestion. We have performed PRSS21 staining in the testis and show that the PRSS21 protein level is not decreased in late spermatids, but its enrichment in the CD is compromised (**Fig. R3-2**). We have added this new panel to **Fig. S9** of the revised manuscript.

Fig. R3-2. SYPL1 deficiency leads to the loss of PRSS21 in the CD of step 16 spermatids. Immunofluorescence showing the absence of PRSS21 in CDs of step 16 spermatids compared to the HET sperm (yellow arrows indicating positive signal in CDs), scale bar, 30 μ m.

5. The authors should determine the phosphorylation status in WT and KO sperm under capacitating and non-capacitating conditions.

Response: We appreciate this suggestion. However, our major finding is a SYPL1-mediated vesicle pathway for CD formation during spermiogenesis in the testis. A phosphoproteome study on sperm in the epididymis is a tremendous undertaking requiring a solid hypothesis. Even if the protein phosphorylation levels of the KO sperm changed in the capacitation and non-capacitation status compared to the control, it may be a secondary effect of SYPL1 deficiency after sperm release into the epididymis. Importantly, SYPL1 is a transmembrane protein that functions as a vesicle regulator rather than a kinase. The analysis of protein phosphorylation levels in the capacitation and non-capacitation conditions is not directly relevant to the function SYPL1 and we believe is beyond the scope of this manuscript.

Minor concerns:

1. Fig. 1d: PRSS21 (36.2 kD) is most abundant in testis; the other faint bands are probably non-specific binding. Is there a RT-PCR signal in tissues other than testis?

Response: We believe the reviewer meant SYPL1 rather than PRSS21 in Fig. 1d. We have performed RT-PCR experiments in different tissues as suggested and the results are shown below (**Fig. R3-3**).

Fig. R3-3. *Sypl1* mRNA expression in multiple tissues by RT-PCR.

2. Fig. 1f: Please clarify the sperm origin (testis vs. epididymis) and provide a brightfield image for localization.

Response: **Fig. 1f** shows isolated sperm from the cauda epididymis, we have added this information in the figure legend. We have added a bright field alone image to the new **Fig. 1f**.

3. Fig. 2f-g, S2f, S3: Please include scale bars.

Response: Scale bars have been included in **Fig. 2f-g, S2f, S3** in the revised manuscript. Thank you.

Literature cited

Banerjee, M., Joshi, S., Zhang, J., Moncman, C.L., Yadav, S., Bouchard, B.A., Storrie, B., and Whiteheart, S.W. (2017). Cellubrevin/vesicle-associated membrane protein-3-mediated endocytosis and trafficking regulate platelet functions. *Blood* *130*, 2872-2883.

Barrachina, F., Battistone, M.A., Castillo, J., Mallofre, C., Jodar, M., Breton, S., and Oliva, R. (2022). Sperm acquire epididymis-derived proteins through epididymosomes. *Hum Reprod* *37*, 651-668.

Borisovska, M., Zhao, Y., Tsytsyura, Y., Glyvuk, N., Takamori, S., Matti, U., Rettig, J., Sudhof, T., and Bruns, D. (2005). v-SNAREs control exocytosis of vesicles from priming to fusion. *EMBO J* *24*, 2114-2126.

Eshkind, L.G., and Leube, R.E. (1995). Mice lacking synaptophysin reproduce and form typical synaptic vesicles. *Cell Tissue Res* *282*, 423-433.

Gordon, D.E., Chia, J., Jayawardena, K., Antrobus, R., Bard, F., and Peden, A.A. (2017). VAMP3/Syb and YKT6 are required for the fusion of constitutive secretory carriers with the plasma membrane. *PLoS genetics* *13*, e1006698.

Haass, N.K., Kartenbeck, M.A., and Leube, R.E. (1996). Pantophysin is a ubiquitously expressed

synaptophysin homologue and defines constitutive transport vesicles. *J Cell Biol* *134*, 731-746.

Hu, C., Hardee, D., and Minnear, F. (2007). Membrane fusion by VAMP3 and plasma membrane t-SNAREs. *Experimental cell research* *313*, 3198-3209.

McMahon, H.T., Bolshakov, V.Y., Janz, R., Hammer, R.E., Siegelbaum, S.A., and Sudhof, T.C. (1996). Synaptophysin, a major synaptic vesicle protein, is not essential for neurotransmitter release. *Proc Natl Acad Sci U S A* *93*, 4760-4764.

Nozawa, T., Minowa-Nozawa, A., Aikawa, C., and Nakagawa, I. (2017). The STX6-VTI1B-VAMP3 complex facilitates xenophagy by regulating the fusion between recycling endosomes and autophagosomes. *Autophagy* *13*, 57-69.

Pfeffer, S.R. (2013). A prize for membrane magic. *Cell* *155*, 1203-1206.

Schraw, T.D., Rutledge, T.W., Crawford, G.L., Bernstein, A.M., Kalen, A.L., Pessin, J.E., and Whiteheart, S.W. (2003). Granule stores from cellubrevin/VAMP-3 null mouse platelets exhibit normal stimulus-induced release. *Blood* *102*, 1716-1722.

Sharma, U., Sun, F., Conine, C.C., Reichholf, B., Kukreja, S., Herzog, V.A., Ameres, S.L., and Rando, O.J. (2018). Small RNAs Are Trafficked from the Epididymis to Developing Mammalian Sperm. *Dev Cell* *46*, 481-494 e486.

Yang, C., Mora, S., Ryder, J.W., Coker, K.J., Hansen, P., Allen, L.A., and Pessin, J.E. (2001). VAMP3 null mice display normal constitutive, insulin- and exercise-regulated vesicle trafficking. *Molecular and cellular biology* *21*, 1573-1580.

REVIEWERS' COMMENTS

Reviewer #1 (Remarks to the Author):

This stellar group of authors have provided a thoughtful response to the review, with revisions including new data as well as requested analyses. These changes have made a very good manuscript even better and more impactful.

One small edit:

Line 130 – the word “sterile” (implying no gametes) should be changed to “infertile” (implying no offspring)

Reviewer #2 (Remarks to the Author):

My concerns have been adequately addressed. Very nice work.

Reviewer #3 (Remarks to the Author):

My concerns have been addressed in the revised manuscript.

A point-by-point response to the reviewers' comments

REVIEWERS' COMMENTS

Reviewer #1 (Remarks to the Author):

This stellar group of authors have provided a thoughtful response to the review, with revisions including new data as well as requested analyses. These changes have made a very good manuscript even better and more impactful.

One small edit:

Line 130 – the word “sterile” (implying no gametes) should be changed to “infertile” (implying no offspring)

Response: We thank the reviewer for this suggestion and have changed it to “infertile”.

Reviewer #2 (Remarks to the Author):

My concerns have been adequately addressed. Very nice work.

Reviewer #3 (Remarks to the Author):

My concerns have been addressed in the revised manuscript.